# Role of Sex and Early Life Stress Experience on Porcine Cardiac and Brain Tissue Expression of the Oxytocin and H_2_S Systems

**DOI:** 10.3390/biom14111385

**Published:** 2024-10-30

**Authors:** Franziska Münz, Nadja Abele, Fabian Zink, Eva-Maria Wolfschmitt, Melanie Hogg, Claus Barck, Josef Anetzberger, Andrea Hoffmann, Michael Gröger, Enrico Calzia, Christiane Waller, Peter Radermacher, Tamara Merz

**Affiliations:** 1Institute for Anesthesiological Pathophysiology and Process Engineering, University Ulm, 89081 Ulm, Germanypeter.radermacher@uni-ulm.de (P.R.); tamara.merz@uni-ulm.de (T.M.); 2Department of Anesthesiology and Intensive Care Medicine, University Hospital Ulm, 89081 Ulm, Germany; 3Department of Psychosomatic Medicine and Psychotherapy, General Hospital Nuremberg, 90419 Nuremberg, Germany

**Keywords:** oxytocin, oxytocin receptor, cystathionine-β-synthase, cystathionine-γ-lyase, prefrontal cortex, hypothalamus

## Abstract

Early life stress (ELS) significantly increases the risk of chronic cardiovascular diseases and may cause neuroinflammation. This post hoc study, based on the material available from a previous study showing elevated “serum brain injury markers” in male control animals, examines the effect of sex and/or ELS on the cerebral and cardiac expression of the H_2_S and oxytocin systems. Following approval by the Regional Council of Tübingen, a randomized controlled study was conducted on 12 sexually mature, uncastrated German Large White swine of both sexes. The control animals were separated from their mothers at 28–35 days, while the ELS group was separated at day 21. At 20–24 weeks, animals underwent anesthesia, ventilation, and surgical instrumentation. An immunohistochemical analysis of oxytocin, its receptor, and the H_2_S-producing enzymes cystathionine-β-synthase and cystathionine-γ-lyase was performed on hypothalamic, prefrontal cortex, and myocardial tissue samples. Data are expressed as the % of positive tissue staining, and differences between groups were tested using a two-way ANOVA. The results showed no significant differences in the oxytocin and H_2_S systems between groups; however, sex influenced the oxytocin system, and ELS affected the oxytocin and H_2_S systems in a sex-specific manner. No immunohistochemical correlate to the elevated “serum brain injury markers” in male controls was identified.

## 1. Introduction

Experiences of early life stress (ELS) are associated with an increased incidence and severity of chronic diseases, including cardiovascular diseases [1,2,3,4]. This effect was reported to be more pronounced in women [5,6,7], contrasting with the higher overall incidence of these diseases in the male population [8,9]. The indication of greater vulnerability to ELS in women is supported by experimental data from a model of early life adversity in pigs: female subjects of the same age and weight exhibited significantly more pronounced damage in the gastrointestinal mucosa compared to their male counterparts [10,11].

Oxytocin (OT), a neuropeptide synthesized in the hypothalamus, has been extensively studied for its involvement in cardiovascular function, social bonding, and stress regulation. It exerts its effects through the oxytocin receptor (OT-R). OT-R is expressed not only in the brain but also in other organs, including the heart [12]. It also has cardioprotective properties by enhancing glucose utilization [13,14], stimulating the nitric oxide system [15,16], and exerting a negative chronotropic effect [15]. The expression of OT-R can be influenced by both sex and/or ELS, e.g., female patients with traumatic childhood experiences (“Childhood Maltreatment”) exhibited a reduced expression of OT-R in PBMCs [17]. However, the available data on a possible impact of sex on the expression of OT-R is inconclusive, with reports of higher OT-R expression in males compared to females in mice [18], unchanged OT-R expression in humans [19], or even diminished OT-R binding in males compared to females in prairie voles [20], suggesting species-specific sex differences in OT-R expression in the brain [21].

Hydrogen sulfide (H_2_S) is referred to as an endogenous signaling molecule with implications in cardiovascular and nervous system physiology. It is synthesized by enzymes, e.g., cystathionine-β-synthase (CBS) and cystathionine-γ-lyase (CSE), and thus plays a regulatory role in various cellular functions including the vasodilation, neurotransmission, and modulation of the inflammatory responses [22]. Furthermore, H_2_S mediates cardioprotection through its anti-oxidant properties [22] and its effects on cardiac mitochondrial function [23,24]. The expression of CSE can be affected by ELS in a severity-dependent way: e.g., “chronically” stressed male mice expressed less myocardial CSE compared to controls [25]. However, the putative effects of ELS on the expression of CBS, and in this context, particularly sex-dependent ones on the expression of CSE and CBS, remain largely unknown.

Both OT-R and CSE are expressed in similar cell types in the cardiovascular system, i.e., cardiomyocytes and endothelial cells [26,27], and have parallel effects in blood pressure regulation and the mediation of cardiac contractility, as well as cardioprotective and anti-oxidant effects [28]. Thus, it is tempting to speculate that the OT/OT-R and H_2_S systems interact in response to both physical and psychological trauma [29]. Physical trauma typically represents a challenge for the cardiovascular system, affecting blood pressure and fluid balance. The hypothalamus plays a central role in regulating blood and body fluid volume and osmolality [30]. The paraventricular nuclei (PVN), as the location of interaction between the H_2_S and oxytocin systems in maintaining fluid homeostasis, assume particular importance in this context [31]: in rats, water deprivation increased the sulfide concentrations in the medial basal hypothalamus, whereas intracerebroventricular injection of the sulfide-releasing salt Na_2_S not only had the opposite effect, but also increased plasma OT concentrations.

In the intervention study underlying this work, the serum levels of the “brain injury markers” MAP-2, GFAP, NSE, and Protein S100β were significantly higher in male than in female control animals, while no gender-dependent difference in ELS animals was apparent [32]. These findings suggest a gender-specific difference in porcine brain biochemistry at baseline, which, moreover, may be differentially affected by ELS. Moreover, systemic cytokine levels had also shown a sex-specific response depending on the presence/absence of ELS [32]: as reported in the previous publication, TNF-α and IL-10 levels were significantly higher in male control animals than in any of the other groups, whereas IL-6 was not affected. Hence, cytokine levels dropped in males with ELS, but were not affected in females. We speculated that biological effects of stress may be more pronounced in males than females. Together with the effect of ELS on myocardial CSE and OT-R expression observed in mice [25], the present post hoc study of heart and brain tissue specimens available from a porcine ELS model investigated the effect of *both* sex and/or ELS on the cardiac and cerebral (hypothalamic PVN and prefrontal cortex) H_2_S and oxytocin systems using immunohistochemistry. In addition to the PVN, the prefrontal cortex was investigated because it holds significant importance in cognitive processing, behavior, and adaptation to the environment and its plasticity is affected by ELS [33,34].

## 2. Materials and Methods

### 2.1. Animals

This study is a post hoc analysis of material available from the above-mentioned interventional study [32]. The experiments had been conducted following approval by both the University of Ulm Animal Care Committee and the Federal Authorities for Animal Research (Regierungspräsidium Tübingen; Reg.-Nr. 1559, approval 29 October 2021) and adhering to the National Institute of Health Guidelines on the Use of Laboratory Animals and the European Union “Directive 2010/63/EU on the protection of animals used for scientific purposes”. The data presented originate from twelve sexually mature German Large White pigs (median [interquartile range] age 23 [22; 24] weeks, bodyweight 82 [67; 93] kg), evenly distributed by sex (n = 3 males/females per group) (see Figure 1A). Animals assigned to the “control” group were weaned between day 28 and 35 after birth, which is consistent with the standard weaning practices in swine husbandry (see Figure 1B). In contrast, those subjected to ELS experienced weaning at day 21 after birth. This time point was selected for several reasons: (*i*) it aligns with the earliest weaning period for swine as stipulated in the Federal German regulations on farm animal husbandry; (*ii*) it has been referred to comply with animal welfare standards, as assessed in the chapter no. 90 entitled “Influence of weaning age on piglet behavior” of the report on “Environmentally compatible and site-specific agriculture” by the Rheinische Friedrich-Wilhelms-University, Bonn, Germany; and (*iii*) it aims to circumvent pathological clinical manifestations associated with earlier weaning (at day 16–18) observed in other ELS models, e.g., diarrhea, weight loss, and/or dysfunction of the intestinal mucosal barrier [35,36,37,38]. In order to mitigate inter-individual variations pertaining to age and developmental stage, an effort was made to select every two pairs of control and ELS animals from the same litter.

### 2.2. Anesthesia and Surgery

Anesthesia and surgical procedures have been described in detail in a previously published study [32]. The experimental timeline is depicted in Figure 1C. On the morning of the experimental day, pigs received intramuscular pre-medication consisting of 2 mg/kg of azaperone and 0.5–1 mg/kg of midazolam, followed by the insertion of a peripheral venous catheter in an ear vein. General anesthesia was then induced using propofol (1.5–2 mg/kg) and ketamine (1 mg/kg), with subsequent endotracheal intubation and administration of fentanyl (20 µg/kg). Muscle paralysis was achieved via pancuronium (0.1 mg/kg). Mechanical ventilation was initiated with the following parameters: tidal volume set at 8 mL/kg, respiratory rate adjusted to 8–12 breaths/minute to achieve an arterial PCO_2_ (P_a_CO_2_) of 35–40 mmHg, inspiratory/expiratory ratio (I/E) of 1:1.5, fraction of inspiratory oxygen (F_i_O_2_) set at 0.3, and positive end-expiratory pressure of 10 cm H_2_O to prevent atelectasis formation [32]. Anesthesia was sustained through continuous intravenous infusion of propofol at a rate of 10 mg/kg/h. To maintain fluid homeostasis, a balanced electrolyte solution (10 mL/kg/h, Jonosteril 1/1^®^, Fresenius Kabi, Bad Homburg, Germany) was administered. A 9F-metal-sheathed catheter (Arrow^®^ International Inc. (Teleflex), Morrisville, NC, USA) was surgically inserted into the left iliac artery to enable continuous monitoring of blood pressure and blood sampling. Following the completion of surgical instrumentation, adjustments were made to the ventilator settings to achieve an inspiratory/expiratory (I/E) ratio of 1:2, F_i_O_2_ set at 0.21, and zero end-expiratory pressure (0 cmH_2_O) in order to closely replicate physiological conditions.

### 2.3. Experimental Protocol

All experiments adhered to a rigorous timeline to mitigate the influence of circadian rhythm. Specifically, intramuscular pre-medication was consistently administered at 06:00 h, induction of general anesthesia commenced at 07:00 h, and subsequent surgical instrumentation lasted approximately 45 min. As outlined in the prior publication [32], arterial blood sampling occurred one hour after completion of surgical instrumentation, immediately preceding euthanization via KCl injection after further deepening of anesthesia. Immediately post-mortem, the heart was excised. Thereafter, the head of the pig was severed. A midline incision was made on the forehead, and the skin and muscle tissue covering the skull were dissected away. Using a bone saw and chisel, the top portion of the skull was removed. Carefully, the dura mater was incised with a scalpel. Subsequently, the head of the pig was positioned to facilitate the disconnection of cranial nerves, resulting in the release of the brain from the skull. The brain was then gently collected by hand. Without delay, brain tissue was immersed in formalin for preservation. Notably, the duration of individual experiments did not deviate by more than 15 min.

### 2.4. Immunohistochemistry

Immunohistochemical analysis was employed to assess the myocardial and cerebral expression levels of OT and its receptor, alongside the principal H_2_S-producing enzymes CBS and CSE. Immunohistochemistry was chosen for several reasons: (*i*) it is widely acknowledged in the literature that densitometric analysis of colorimetric immunohistochemical staining provides comparable reliability to Western blotting for quantifying protein levels [39]; (*ii*) significant correlations have been observed between densitometric values and those obtained from Western blotting analysis [40]; and (*iii*) in contrast to Western blotting, immunohistochemical analysis of tissue enables the identification of spatial distribution and protein expression in specific cell types within the tissue sample.

All brain specimens were subjected to consistent fixation in a 4% formalin solution for a duration of 6 days. Subsequently, each brain underwent dissection into sequential coronal sections with a thickness of 4 mm, extending from the frontal to occipital regions. In instances where the size of the macroscopic section exceeded the dimensions of the embedding cassette (26 × 3 × 4 mm), it was horizontally aligned and further subdivided into a maximum of five pieces. This dissection process was meticulously executed to ensure the preservation of tissue integrity and to facilitate subsequent reconstruction of the entire section for analytical purposes. For this study, the macroscopic sections, which included the hypothalamus and the prefrontal cortex, were selected for thorough examination.

Immunohistochemistry of cardiac specimens was conducted following established procedures [41]. Left ventricular cardiac samples obtained immediately post-mortem were fixed in formalin (4%) for 6 days.

Brain and myocardial tissue specimens underwent dehydration and were embedded in paraffin blocks. Sections measuring 3–5 μm in thickness underwent deparaffinization using xylene, followed by rehydration through a sequential series of ethanol and deionized water. Immunohistochemistry was performed following the methodology outlined previously [42]. Briefly, after deparaffinization, heat-induced antigen retrieval was conducted in citrate solution (pH 6). Subsequently, blocking was carried out using 10% normal goat serum (Jackson ImmunoResearch Laboratories, Inc., West Grove, PA, UK) prior to incubation with the following primary antibodies: H_2_S-producing enzymes anti-CBS (Protein Tech, Manchester, UK, 14787-1-AP, RRID: AB_2070970), anti-CSE (Protein Tech, 12217-1-AP, RRID: AB_2087497), anti-OT (Millipore, Taufkirchen, Germany, AB911, RRID: AB_2157629), and anti-OT-R (Protein Tech, 2304523045-1-AP, RRID: AB_2827435). The optimal dilution for all primary antibodies was adjusted according to the manufacturer’s recommendations and titrated to optimal concentrations for the samples at hand (Table 1). Primary antibody detection was executed utilizing the Dako REAL detection system (anti-mouse, anti-rabbit, alkaline phosphatase-conjugated). Visualization was achieved using red chromogen (Dako REAL; Dako, Agilent Technologies, Santa Clara, CA, USA), followed by counterstaining with hematoxylin (Sigma, St. Louis, MO, USA). The slides were examined using a Zeiss Axio Imager A1 microscope equipped with a 10× and 40× objective lens. Quantitative analysis was conducted on 800,000 μm^2^ sections utilizing the Zen Image Analysis Software (Zeiss, Oberkochen, Germany). The findings are presented as the percentage of positively stained area relative to the total area [43].

### 2.5. Statistical Analysis

Due to the very limited data with the early weaning pig model itself and the fact that we were the first to analyze the effects of early weaning on the pig brain in particular, there was no basis of preliminary data and/or relevant previous studies to allow for a power calculation. Thus, the local Animal Care Committee and the Federal Authorities for animal protection considered this experiment as a pilot study, and, consequently, determined a maximum n = 12 for that type of study. Statistical analysis was performed using GraphPad Prism Version 8. Intergroup disparities were evaluated employing a two-way ANOVA. Given the small number of animals per group, no testing for outliers has been performed.

## 3. Results

Figure 2 shows representative images (left side) and the quantitative analysis (right panels) of the myocardial tissue detection of the H_2_S-producing enzyme CSE (upper graphs) and OT-R (lower graphs). Neither myocardial CSE expression nor OT-R showed any significant intergroup differences, regardless of the presence or absence of ELS or sex. However, there is no overlap in myocardial OT-R expression between the male control and ELS animals, respectively.

Figure 3 and Figure 4 present representative images (left panels) and the quantitative analysis (right panels) of the prefrontal cortex staining for the H_2_S-producing enzyme CBS (Figure 3) in the gray (upper graphs) and white matter (lower panel) and for OT-R in the white matter (Figure 4). The expression of OT-R was depicted exclusively in the white matter, as OT-R expression was nearly absent in the gray matter. Neither CBS in the gray and white matter, nor OT-R in the white matter, showed any significant intergroup differences. As in the myocardium (Figure 2), there is no overlap in the expression of OT-R in the PFC between the male control and ELS animals either. CSE and OT were nearly absent in the prefrontal cortex.

Representative images (left panels) and the quantitative analysis (right panels) of the paraventricular nucleus of the hypothalamus staining for OT (upper graphs), its receptor (bottom panels), the H_2_S-producing enzymes CBS (upper panels), and CSE (bottom graphs) are presented in Figure 5 and Figure 6. None of these target proteins exhibited any significant intergroup variance, irrespective of sex or the presence or absence of ELS. Nevertheless, male animals exhibited higher OT expression compared to females. Additionally, the expression of the OT-R was greater in male controls than in females.

## 4. Discussion

The aim of this investigation was to assess the effects of sex and early weaning as a model of ELS on the myocardial protein levels of CSE and OT-R, the protein levels of CBS and OT-R in the prefrontal cortex and of OT, OT-R, CBS, and CSE in the paraventricular nucleus of the hypothalamus. The main findings of the present study were (*i*) that none of these parameters showed any significant differences in expression, (*ii*) sex influenced the expression of OT and its receptor, and (*iii*) stressed males exhibited a trend towards reduced expression of myocardial CSE, OT-R, and cerebral CBS and OT-R when compared to control males.

Neither sex nor ELS experiences resulted in significant differences between groups. However, we observed that the myocardial expression of CSE tended to be lower in the stressed male animals when compared to the male controls, an effect that was not observed in the female animals. These results align with findings from a study using a mouse model, which demonstrated a reduced myocardial CSE expression in male subjects with “chronic” long-term separation stress when compared to male control mice [25]. In that study, ELS was also induced by maternal separation. However, unlike in our present study, two distinct stress protocols were implemented: the “chronic” stress group underwent earlier and more prolonged maternal separation when compared to the “mild” stress group. Thus, it was demonstrated that the effects on myocardial CSE expression are markedly influenced by the type of stress and/or severity. In fact, the aforementioned study showed that reduced CSE expression occurred exclusively in “chronically” stressed animals, contrasting with mice subjected to the “mild” stress protocol. Another pivotal distinction is that only male experimental animals were employed, thus precluding any extrapolation to the female population.

Given the interaction between the H_2_S system and the oxytocin system [29], we also conducted a more detailed analysis of the OT/OT-R expression, and we did not detect any significant differences in myocardial OT-R expression based on sex or ELS experience. However, similar to CSE, a markedly reduced myocardial OT-R expression was found in the male animals suffering from ELS, further supporting sex-specific effects of ELS in our model. In the above-mentioned murine study, a diminished myocardial OT-R expression in male subjects with “chronic” long-term separation stress compared to male control mice was also observed [25]. Thus, we confirmed the previously published data from murine ELS here in our porcine model. Consistent with these experimental findings, a reduced expression of OT-R in peripheral blood mononuclear cells (PBMCs) has been observed in patients who experienced early life stress [44].

In addition to psychological trauma, physical trauma also appears to lead to a reduction in myocardial OT-R expression. This was demonstrated in mice of both sexes following exposure to combined cigarette smoke and acute blunt chest trauma [45]. A decreased OT-R expression in infarcted left ventricular tissue was directly related to worsened myocardial injury [46], underscoring the detrimental impact of reduced OT-R expression on cardiovascular health. Similarly, a lack of CSE is associated with an aggravation of myocardial fibrosis and heart failure in a murine model [47], suggesting that CSE plays a crucial role for proper cardiac function. The dysregulation of CSE and OT-R in the myocardium and associated interference with cardioprotective mechanisms of individuals experiencing ELS might mediate the higher risk for ELS-associated cardiovascular disease in adults on a molecular level. In fact, in a porcine resuscitated model of septic shock with underlying atherosclerosis, we were able to show that atherosclerosis reduced coronary artery [48], and myocardial CSE expression [49] was associated with reduced cardiac output in response to sepsis [49], at a similar degree to that observed in human atherosclerotic septic patients compared to cardiovascularly healthy septic patients [50].

Sex-specific effects of ELS in the expression of OT-R were identified not only in the myocardium but also in the white matter of the prefrontal cortex. Similar to the myocardium, stressed males expressed less OT-R compared to control males. Our findings corroborate the observations from a previously published study in rats [51]. Following “Long-Term Neonatal Maternal Separation”, male rats exhibited a decreased expression of OT-R in the prefrontal cortex when compared to their control counterparts. Sex-specific differences could not be inferred from this study, as it focused exclusively on male rats. However, another study involving rats, both sex-specific differences and the impact of ELS (in this case swim, restraint, and elevated platform stress) were investigated [52]. After ELS experience, neither male nor female animals exhibited any alteration of the OT-R expression in the prefrontal cortex [52]. However, a sex-specific difference was evident: females generally expressed higher levels of OT-R in the prefrontal cortex than males [52].

As previously noted in the context of myocardial expression involving CSE and OT-R, the interaction between the oxytocin system and the H_2_S system is also observed in the white matter of the prefrontal cortex. While the already low expression of CBS in male control animals further decreased in stressed males and became almost undetectable, female pigs not only express more CBS, but are also unaffected by ELS experiences. To the best of our knowledge, there are currently no available data in the literature regarding the influence of sex or ELS on the protein levels of CBS in the prefrontal cortex, which makes it difficult to interpret these findings, especially given the lack of significance due to the low number of animals in our study.

Moreover, the H_2_S and oxytocin systems have been scrutinized in greater detail within the paraventricular nucleus of the hypothalamus, a pivotal center influencing the interaction of these two systems especially under stress conditions [53]. Clearly, no significant intergroup differences were detected in either the expression of OT or its receptor, irrespective of sex and/or the absence/presence of ELS experience. Nevertheless, male animals exhibited higher OT expression than females. Furthermore, the expression of OT-R tended to be greater in male controls than in females. A sex-specific difference was also observed, with a notable decrease in OT-R expression in stressed males, while expression levels remained unchanged in stressed females. Variable effects of stress on the oxytocin system have been reported in previous studies, inasmuch as decreased [54] or unchanged [52] OT expression in the male hypothalamus were reported. In rodents, inadequate maternal care was shown to reduce both hypothalamic and blood plasma OT and OT-R expression in females [55]. However, maternal separation altered both the OT expression and OT-R binding in an age- and sex-specific manner [56,57]. These at least partially divergent findings could be attributed, on the one hand, to the varying types of stressors employed, and, on the other hand, to the differences in the animal models utilized. The pig, for example, is considered a highly pertinent translational model owing to its structural resemblance to the human brain. Features such as gyri and sulci (gyrencephalic brain), a white matter to gray matter ratio similar to that of humans, and the presence of a tentorium cerebelli distinguish it from the rodent brain, thereby mirroring more closely human pathophysiology [58,59]. In this context, our findings of a sex-specific OT-R expression in the male hypothalamus agree well with previous findings in males [52]. ELS-induced changes in hypothalamic OT/OT-R, CSE, and CBS expression might reflect the impaired hypothalamic regulatory capacity of blood volume after fluid shifts.

Despite the known interaction between the oxytocin and H_2_S systems, no significant intergroup differences in CBS and CSE expression in the hypothalamus were detected, irrespective of sex and/or ELS experience. Again, to the best of our knowledge, there are no existing data in the literature that address the impact of sex or ELS on the protein levels of CBS and/or CSE in the hypothalamus. However, when comparing the results with physical trauma, notable distinctions emerge. A previously published study has demonstrated that following traumatic brain injury, the expression of CBS and CSE in the hypothalamus decreases [60]. In contrast to our study, rats were used, exclusively males, precluding any sex-specific conclusions.

There are some limitations to this study. Our study is limited by the fact that, for this pilot experiment, we received approval from the Animal Care Committee of Universität Ulm and the Federal Authorities for Animal Research (Regierungspräsidium Tübingen) for only six control and six ELS animals. As a result, conducting a power calculation was not possible. The limited number of animals in each group may have resulted in potentially overlooking significant differences.

The NLRP3 inflammasome is a central effector in neuroinflammation [61,62], which has not been investigated in the present study. Since it can be inhibited by both OT-signaling [63] and H_2_S [64], which might play a role in depression and acute brain injury, respectively, it is a relevant target for future investigations.

## 5. Conclusions

In this post hoc analysis, no immunohistochemical correlate to the elevated serum concentrations of the “brain injury markers” in the male control animals could be identified. The trends towards lower levels of CSE and OT-R in myocardial tissue of stressed males support previous data on the effect and interaction of CSE and OT-R in the heart after physical and psychological stress [25,45] and might represent a molecular correlate for increased cardiovascular risk in adults with ELS experience.

## Figures and Tables

**Figure 1 biomolecules-14-01385-f001:**
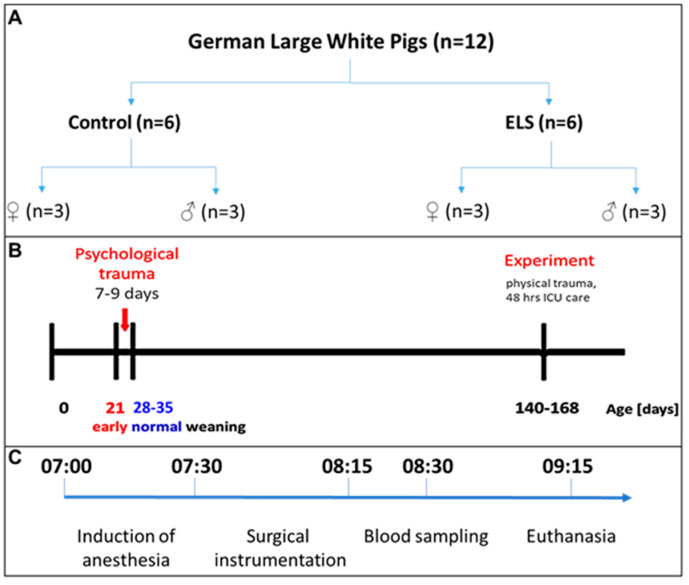
Animals and experimental protocol. (**A**) 12 German Large White pigs were distributed equally into control and early life stress (ELS) with each group containing 3 male non-castrated and 3 female pigs. (**B**) Psychological trauma was induced by early weaning (at postnatal day 21 instead of postnatal day 28–35 for controls). Experiments were performed at postnatal day 140–168. (**C**) Experimental timeline.

**Figure 2 biomolecules-14-01385-f002:**
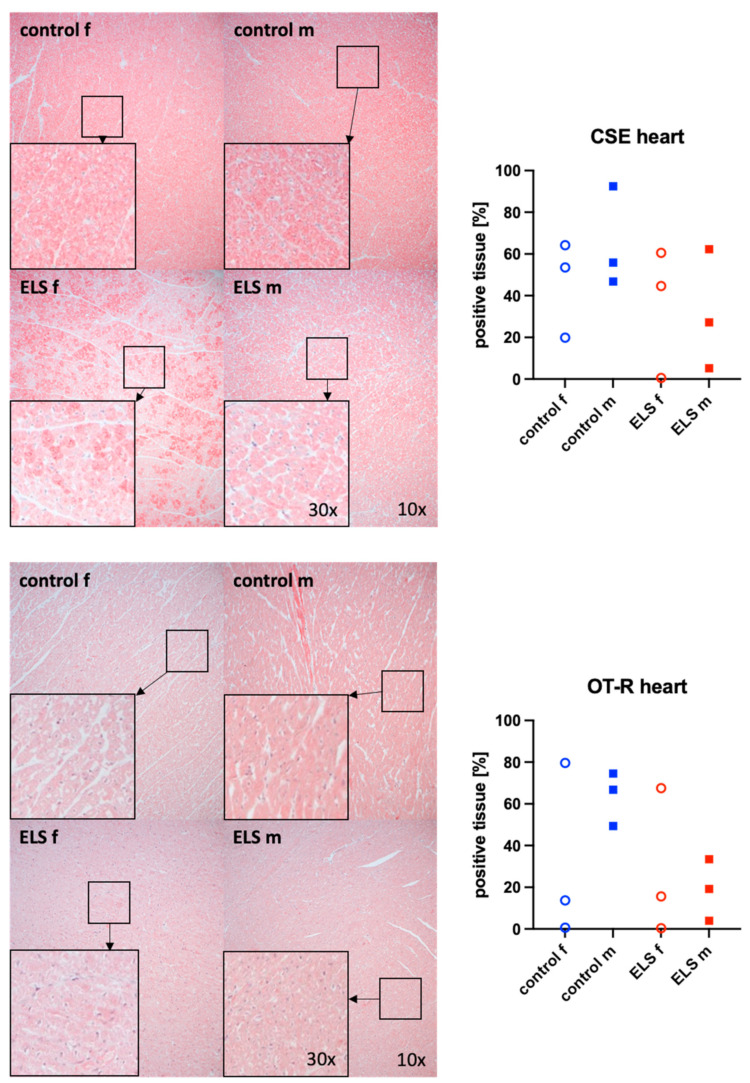
H_2_S-producing enzyme cystathionine-γ-lyase (CSE) (**upper panel**, n = 12) and oxytocin receptor (OT-R) (**lower panel**, n = 12) in the heart (left ventricle). Representative images (**left panel**), 30× magnified images originating from the black box in the 10× magnified image and quantification of immunohistochemical staining as positive tissue [pink, %] (**right panel**). f: female (open circles), m: male (solid squares), ELS: early life stress (red symbols), control (blue symbols).

**Figure 3 biomolecules-14-01385-f003:**
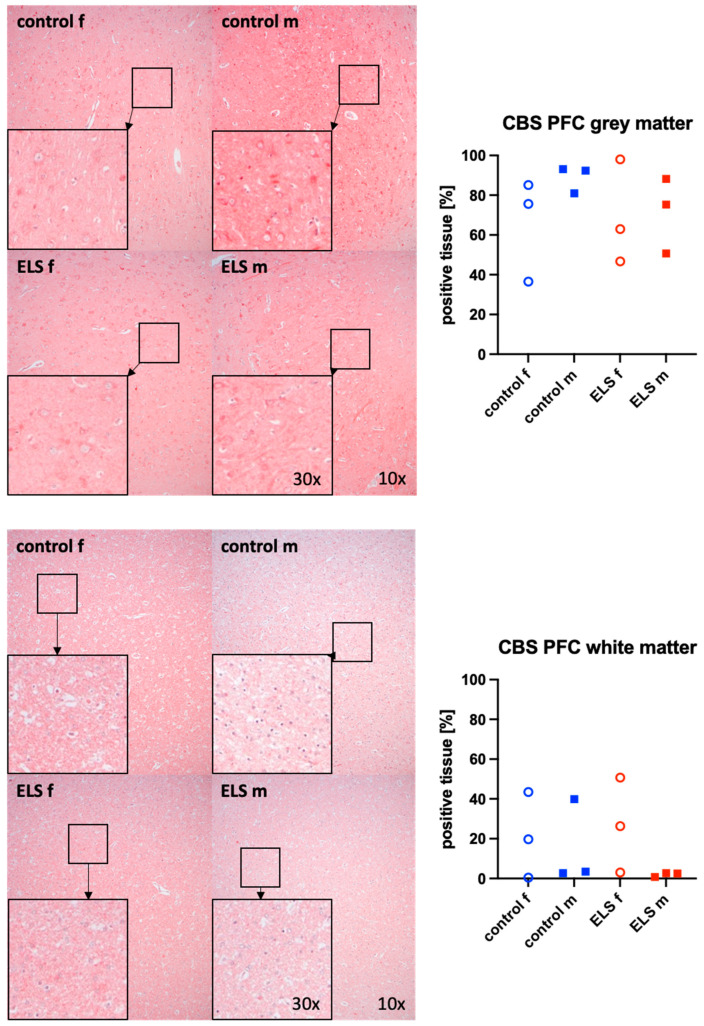
H_2_S-producing enzyme cystathionine-β-synthase (CBS) in the gray matter (**upper panel**, n = 12) and white matter (**lower panel**, n = 12) of the prefrontal cortex. Representative images (**left panel**), 30× magnified images originating from the black box in the 10× magnified image and quantification of immunohistochemical staining as positive tissue [pink, %] (**right panel**). f: female (open circles), m: male (solid squares), ELS: early life stress (red symbols), control (blue symbols).

**Figure 4 biomolecules-14-01385-f004:**
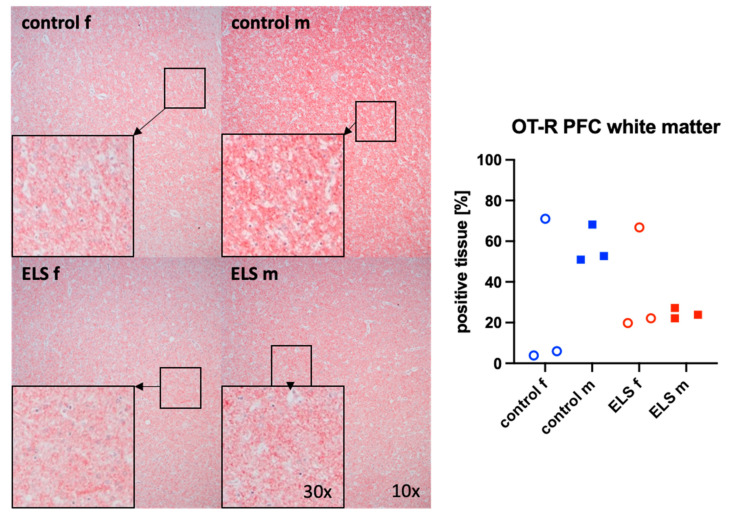
Oxytocin receptor (OT-R) (n = 12) in the white matter of the prefrontal cortex. Representative images (**left panel**), 30× magnified images originating from the black box in the 10× magnified image and quantification of immunohistochemical staining as positive tissue [pink, %] (**right panel**). f: female (open circles), m: male (solid squares), ELS: early life stress (red symbols), control (blue symbols).

**Figure 5 biomolecules-14-01385-f005:**
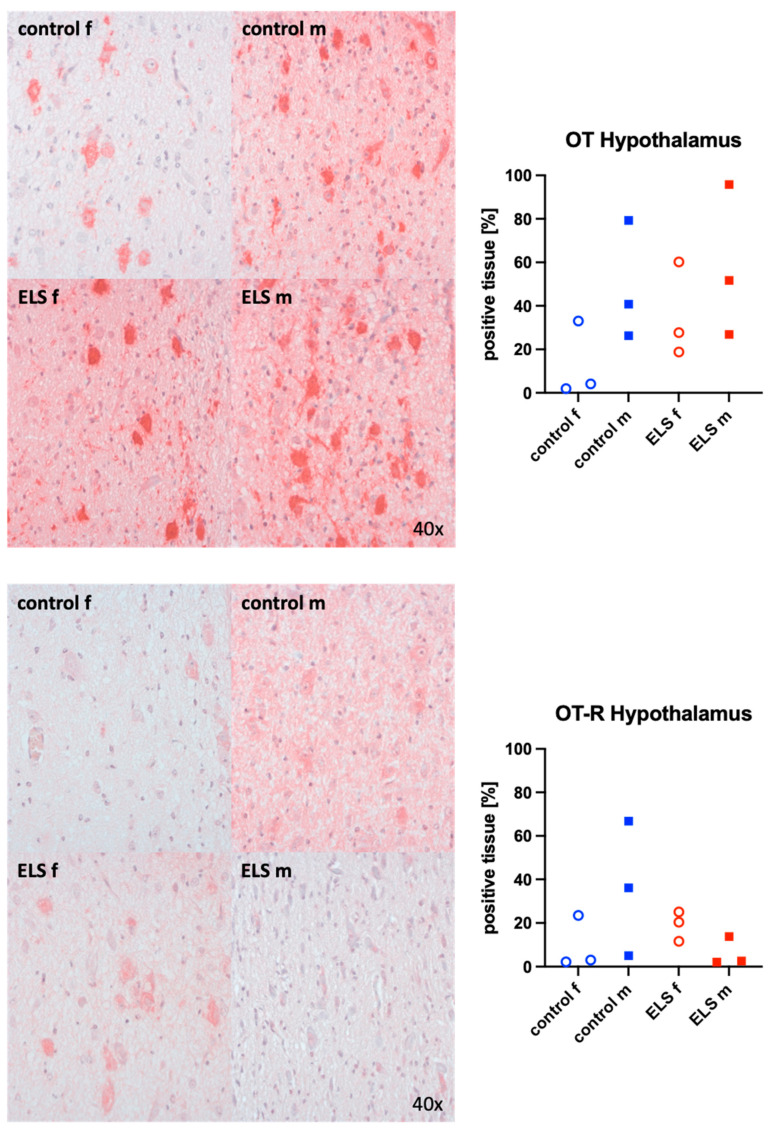
Oxytocin (OT) (**upper panel**, n = 12) and its receptor (OT-R) (**lower panel**, n = 12) in the paraventricular nucleus of the hypothalamus. Representative images (**left panel**) and quantification of immunohistochemical staining as positive tissue [pink, %] (**right panel**). f: female (open circles), m: male (solid squares), ELS: early life stress (red symbols), control (blue symbols).

**Figure 6 biomolecules-14-01385-f006:**
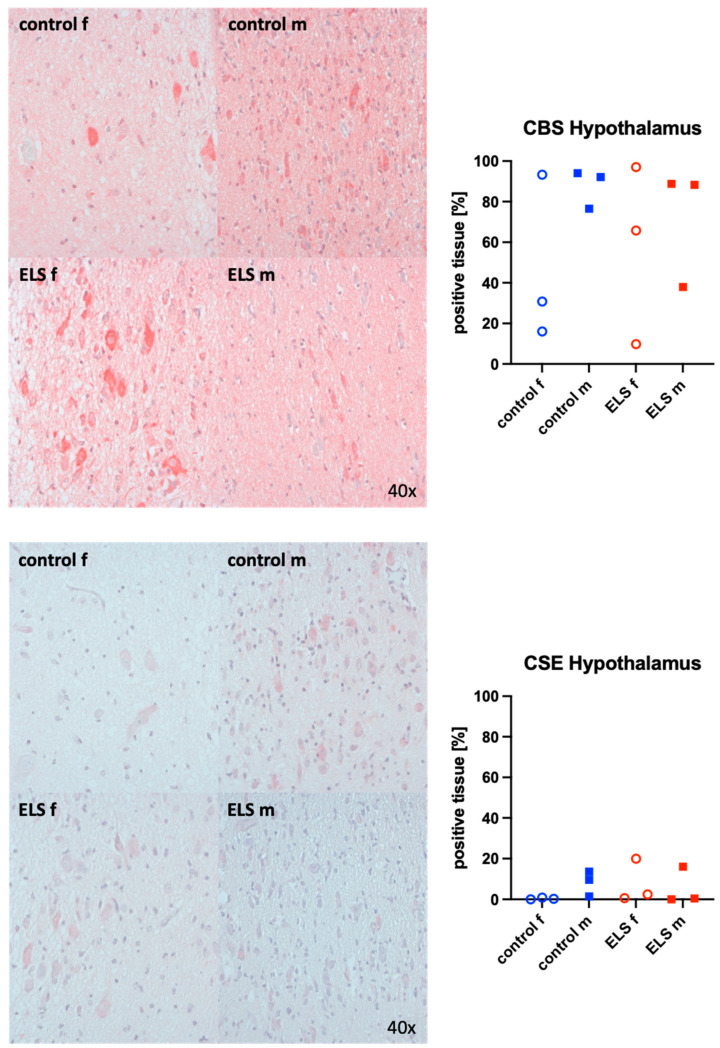
H_2_S-producing enzymes cystathionine-β-synthase (CBS) (**upper panel**, n = 12) and cystathionine-γ-lyase (CSE) (**lower panel**, n = 12) in the paraventricular nucleus of the hypothalamus. Representative images (**left panel**) and quantification of immunohistochemical staining as positive tissue [pink, %] (**right panel**). f: female (open circles), m: male (solid squares), ELS: early life stress (red symbols), control (blue symbols).

**Table 1 biomolecules-14-01385-t001:** Primary antibodies.

Primary Antibody (Source, Catalog No., RRID)	Host Species	Immunogen Sequence	Concentration Used for IHC
**anti-CBS** (Protein Tech, 14787-1-AP, AB_2070970)	Rabbit Polyclonal	CBS fusion protein Ag6437	1:200
**anti-CSE** (Protein Tech, 12217-1-AP, AB_2087497)	Rabbit Polyclonal	Gamma cystathionse fusion protein Ag2872	1:200
**anti-OT** (Millipore, Ab911, AB_2157629)	Rabbit Polyclonal	CYIQNCPLG (Synthetic oxytocin (Sigma) conjugated to thyroglobulin)	1:500
**anti-OT-R** (Protein Tech, 123045-1-AP, AB_2827425)	Rabbit Polyclonal	Oxytocin Receptor fusion protein Ag19074	1:100

## Data Availability

The data presented in this study are available upon request from the corresponding author.

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
