# Peer review of "Role of Sex and Early Life Stress Experience on Porcine Cardiac and Brain Tissue Expression of the Oxytocin and H2S Systems"

_biomolecules, 2024, doi:10.3390/biom14111385_

Round 1
Reviewer 1 Report
Comments and Suggestions for Authors
In this study, Münz and colleagues investigated the impact of sex and/or Early Life Stress (ELS) on the expression of oxytocin (OT) and its receptor (OTR) and H2S-producing enzymes cystathionine-β-synthase (CBS) and cystathionine-γ-lyase (CSE) in the hypothalamic, prefrontal cortex, and myocardial tissue samples, through immunohistochemical analysis. The researchers employed a porcine animal model and established a protocol of ELS in the form of early weaning stress (EWS) at PND 21; in contrast control animals were weaned between PND 28 and 35. This EWS protocol has been reported to induce activation of the hypothalamic-pituitary-adrenal (HPA) axis, upregulation of intestinal corticotropin releasing factor (CRF) and lasting and sex-specific hypersensitivity of secretomotor neuron function and upregulation of the cholinergic ENS.
As stated in the manuscript, a previous study from the same research group published in Frontiers (Münz et al., 2023) has already reported findings from the same animal cohort of 12 German large white swine. Accordingly, as the authors decided to divide their findings into two distinct original articles, this manuscript can be regarded as a follow-up to the previous one. In this reviewer´s opinion, there are several aspects that the authors should address before this manuscript could be considered suitable for publication in this journal.
Introduction
1- Since the aim of the study is to analyze the expression of the OT/ OTR and H2S-producing enzymes CBS and CSE, it would be useful if the authors can elaborate on the rationale behind the decision to investigate these two systems, mentioning previous literature findings which have shown how these networks work in parallel. It would be recommended to state their specific role but also to provide more insight about crucial similarities between OTR and CSE in terms of expression in the same tissue such as cardiomyocytes, endothelial cells and therefore their involvement in blood pressure regulation (Mancardi et al., 20119). Moreover, it would be advisable to mention in the introduction, rather than later in the article, their interaction in the heart and in the brain in response to both physical and psychological trauma (Denoix et al.,2020); Finally, not only their singular properties but also shared cardioprotective, anti-oxidant and anti-inflammatory functions and the evidence that both the OT and H2S systems play gender specific roles (Borland et al., (2019) are recommended to be stated, in order to provide a comprehensive and exhaustive view of these two systems.
LINE 51-55
2- When stating the physiological properties of the H2S system it would be more relevant to mention its specific role in regulating the cardiovascular system.
LINE 60-62
3- The manuscript reports that ‘in the intervention study underlying this paper, the serum levels of the brain markers were elevated in male controls, while no effect of the presence/absence of ELS was apparent’. Subsequently, in the following sentence the authors state directly the objective of this study. This passage lacks coherence. The last part of the introduction would benefit from a more thorough and well-structured discourse, in which the authors elucidate how the objective of their study is developed from previous findings and their own postulated hypothesis.
LINE 143
4- In the immunohistochemistry section, the authors mention the reasons behind choosing the hypothalamus and the prefrontal cortex for their study. It would be recommended to rather include this part in the introduction, so the reader can gain more insight into the role of these two systems in these specific brain regions.
LINE 150
5- Please add literature reference to support your statement.
LINE 238
6- The manuscript reports a decrease of CSE in myocardial tissue in ELS male animals but does not fully explore the mechanistic implications of this change. It would be valuable to include further discussion on how the observed downregulation of this H2s system component might affect cardiovascular physiology.
LINE 252
7- Please state relevant literature bibliography about the interaction between H2S and OT systems.
LINE 255
8- The authors report a reduced myocardial OTR in male animals affected by ELS.
Similarly to the previous CSE finding, these results are not statistically significant. However, it would be recommended to discuss these data in the light of previous papers showing how OTR expression in the heart is directly affected by H2S and vice versa how reduced CSE expression was found in OTR knock out mice. The results of this paper, although not significant, may in part provide further support to previous studies’ results.
LINE 262
9- Please investigate how ELS in the form of early weaning can affect the oxytocinergic system.
LINE 285
10- In this paper it is reported that ‘neither male nor female animals exhibited any alteration of the OTR expression in the PFC. However, a sex specific difference was evident female expressed higher levels of OTR in the PFC compared to male’.
The interpretation of these data should be carefully paraphrased. It is not specified the difference between controls and ELS groups, moreover the difference between male controls and female controls its not higher but rather shows a different distribution.
LINE 288
11-The authors report that CBS in the PFC is decreased in ELS male animals compared to control. In contrast, female animals show similar levels of CBS in both control and ELS group, pointing to a lack of ELS effect in the female gender. Furthermore, it is stated that this finding represents the major result of this research, which hasn’t been reported by any other study in the literature.
This statement should be carefully paraphrased since no statistical significance was achieved. Furthermore, please indicate if a test for outliers (and which) was conducted on the data and state if any data points were excluded or indicate that no test for outliers was conducted. This would allow to evaluate more precisely the result obtained for the control male samples for the CBS PFC expression.
LINE 293
12-The paper reports the results of the OT and OTR expression in the hypothalamus, analyzing the differences in the light of gender and/or ELS effect. The authors point to differences evident in the male gender which shows higher level of OTR in the control group compared to the ELS. Further this result is commented in the light of sex specific differences already shown in other studies and commenting also on the possible divergence due to the use in this study of a porcine animal model. If on one hand it is stated that the pig model represents a highly pertinent translational model due to high resemblance to human brain, no further physiological explanation nor support for the value of this result is provided.
As in the previous findings reported, a thorough discussion of the physiological meaning of these results should be provided.
LINE 315
13-This paper does not report a clear conclusion section. The last paragraph is inconclusive and fragmented, and it does not provide a summarized and detailed reports of the major findings. Rather than comparing the results with previous papers which have major differences in terms of stress an animal model adopted, it would be more advisable to address the results in the light of the physiological and pathophysiological function of the H2S and OT systems first in the hypothalamus, PFC and then in the heart tissue. Therefore, providing realistic interpretation of the findings, which can support previous knowledge reported in other studies.
LINE 326
14- Although in the limitation paragraph it is stated that this study received approval from the Animal Care Committee for only 6 controls and 6 ELS animals, and this has not allowed to conduct a power calculation, still please describe how the number of subjects used for the study was determined (e.g. if based on previous studies of a similar nature *AND* including a sample size calculation/statistical validation, provide relevant citations, or else provide some other verification of sufficient statistical power and validity).
I will be delighted to review again the manuscript after major revision
Author Response
"Please see the attachment."

Reviewer 2 Report
Comments and Suggestions for Authors
In this work the authors investigated the action of early life stress (ELS) in the field of cardiovascular diseases and the consequences in the field of neuroinflammation in an animal model. The authors obtained a series of interesting results in the field of oxytocin, H2S. points. 1- the authors must analyze inflammatory markers such as NLRP3 and related cytokines 2- add a figure relating to the protocol used and the subdivision of the groups 3- add the quantitative expression and activity of the enzymes of H2S 4- comment PMID: 39241894 , PMID: 39042218 5- shorten the discussion
Author Response
"Please see the attachment."

Round 2
Reviewer 1 Report
Comments and Suggestions for Authors
The Authors did a good job in revising the manuscript, which is now acceptable for publication
Reviewer 2 Report
Comments and Suggestions for Authors
I agree